# Determination of Anti-Xa Inhibitor Plasma Concentrations Using a Universal Edoxaban Calibrator

**DOI:** 10.3390/diagnostics13122128

**Published:** 2023-06-20

**Authors:** Annika Burger, Jan-Dirk Studt, Adriana Mendez, Lorenzo Alberio, Pierre Fontana, Walter A. Wuillemin, Adrian Schmidt, Lukas Graf, Bernhard Gerber, Cédric Bovet, Thomas C. Sauter, Nikolaus B. Binder, Michael Nagler

**Affiliations:** 1Department of Clinical Chemistry, Inselspital, Bern University Hospital, University of Bern, 3010 Bern, Switzerland; 2Division of Medical Oncology and Hematology, University Hospital Zurich, University of Zurich, 8006 Zurich, Switzerland; jan-dirk.studt@usz.ch; 3Institute for Laboratory Medicine, Kantonsspital Aarau, 5001 Aarau, Switzerland; adriana.mendez@ksa.ch; 4Service and Central Laboratory of Hematology, Centre Hospitalier Universitaire Vaudois (CHUV), 1011 Lausanne, Switzerland; lorenzo.alberio@chuv.ch; 5Division of Angiology and Hemostasis, Geneva University Hospital, 1205 Geneva, Switzerland; pierre.fontana@hcuge.ch; 6Division of Hematology, Hematology Laboratory, Cantonal Hospital of Lucerne, University of Bern, 3012 Bern, Switzerland; walter.wuillemin@luks.ch; 7Institute of Laboratory Medicine, Clinic of Medical Oncology and Hematology, Municipal City Hospital Zurich Triemli, 8063 Zurich, Switzerland; adrian.schmidt@stadtspital.ch; 8Centre for Laboratory Medicine, 9001 St. Gallen, Switzerland; lukas.graf@zlmsg.ch; 9Clinic of Hematology, Oncology Institute of Southern Switzerland, 6500 Bellinzona, Switzerland; bernhard.gerber@eoc.ch; 10Department of Emergency Medicine, Inselspital, Bern University Hospital, 3010 Bern, Switzerland; thomas.sauter@insel.ch; 11Technoclone Herstellung von Diagnostika und Arzneimitteln GmbH, 1230 Vienna, Austria; nikolaus.binder@technoclone.com

**Keywords:** diagnostic accuracy, anti-Xa assay, laboratory monitoring, rivaroxaban, edoxaban, apixaban, edoxaban, anticoagulants introduction

## Abstract

A universal calibrator for the determination of all anti-Xa inhibitors would support laboratory processes. We aimed to test the clinical performance of an anti-Xa assay utilizing a universal edoxaban calibrator to determine clinically relevant concentrations of all anti-Xa inhibitors. Following a pilot study, we enrolled 553 consecutive patients taking rivaroxaban, edoxaban, or apixaban from nine study centers in a prospective cross-sectional study. The Technochrom^®^ anti-Xa assay was conducted using the Technoview^®^ edoxaban calibrator. Using ultra-high-performance liquid chromatography-tandem mass spectrometry (LC-MS/MS), anti-Xa inhibitor drug concentrations were determined. Sensitivities and specificities to detect three clinically relevant drug concentrations (30 µgL^−1^, 50 µgL^−1^, 100 µgL^−1^) were determined. Overall, 300 patients treated with rivaroxaban, 221 with apixaban, and 32 with edoxaban were included. The overall correlation coefficient (r_s_) was 0.95 (95% CI 0.94, 0.96). An area under the receiver operating characteristic curve of 0.96 for 30 µgL^−1^, 0.98 for 50 µgL^−1^, and 0.99 for 100 µgL^−1^ was found. The sensitivities were 92.3% (95% CI 89.2, 94.6), 92.7% (89.4, 95.1), and 94.8% (91.1, 97.0), respectively (specificities 82.2%, 93.7%, and 94.4%). In conclusion, the clinical performance of a universal, edoxaban-calibrated anti-Xa assay was solid and most drug concentrations were predicted correctly.

## 1. Introduction

A growing population of patients using direct oral anticoagulants (DOAC) are subject to critical situations, including major bleeding, urgent surgery, and planned thrombolysis [1,2,3,4,5,6]. In these situations, the risk of significant bleeding and subsequent complications is high, and relevant drug concentrations will increase this risk even further [7]. Therefore, knowledge of DOAC drug concentrations supports physicians on whether or not reversal agents should be given [8]. Moreover, the time point of surgical or other interventions can be planned [7,8]. Notably, a decision can be made as to whether or not acute thrombolysis can be conducted safely [9,10,11]. Furthermore, the determination of drug levels is also important to detect accumulation in case of renal or hepatic failure or accidental poisoning [12,13]. Furthermore, in specific populations, such as patients with renal impairment, there might potentially be an application to monitor DOAC and adjust dosages [13]. Thus, the accurate and rapid measurement of DOAC drug levels supports the management of patients in various clinical situations [1].

To be useful in daily clinical practice, laboratory tests must be rapidly available. However, standard hemostasis tests such as activated partial thromboplastin time or prothrombin time are not sensitive [14,15]. High-performance liquid chromatography-tandem mass spectrometry (LC-MS/MS) detects DOAC drug levels with high accuracy, but this test is not available in routine clinical practice [16]. Chromogenic anti-Xa assays, commonly used to monitor unfractionated and low-molecular-weight heparin, also demonstrated high accuracy and good reproducibility when measuring rivaroxaban, edoxaban, and apixaban. [17]. However, widespread implementation is hampered by elaborate laboratory procedures: separate calibration and quality controls are necessary for rivaroxaban, edoxaban, and apixaban. A low-molecular-weight heparin calibration was suggested as a universal calibrator to solve this problem [16,17]. Recently, we studied unfractionated heparin as a potential calibrator in this situation [18]. The disadvantage of this approach is that results are not directly given in the correct unit (mg L^−1^). Preliminary data from a pilot study suggest that a universal edoxaban calibration would determine drug levels of all three anti-Xa inhibitors with high accuracy [19].

We conducted a prospective, multicenter study in routine clinical practice to assess the diagnostic accuracy of an anti-Xa assay utilizing a universal edoxaban calibrator to determine critical plasma concentrations of all anti-Xa inhibitors (rivaroxaban, edoxaban, and apixaban). This study therefore represents a proof-of-principle study.

## 2. Methods

### 2.1. Study Design, Setting, and Population

Following a pilot study using spiked samples, we conducted a multicenter cross-sectional study, enrolling consecutive patients from 9 Swiss tertiary hospitals [16,20]. The following inclusion criteria were applied: (a) anticoagulant treatment with either rivaroxaban, edoxaban, or apixaban, (b) drug concentration ordered, (c) age > 18 years, and (d) signed general informed consent if requested by the local authorities. Exclusion criteria were (a) refused informed consent, (b) heparin use, (c) preanalytical problems such as hemolytic samples, (d) more than one drug detected, and (e) not enough sample material. To cover the full range of drug concentrations in clinical practice, samples were collected regardless of the time of last drug consumption. Figure 1 illustrates the study design (CONSORT flow diagram). On all samples, ultra-high-performance liquid chromatography-tandem mass spectrometry (LC-MS/MS) was conducted as a reference standard. [16]. We followed previous recommendations to define the clinically relevant drug concentrations: 30 mgL^−1^, 50 mgL^−1^, and 100 mgL^−1^ [1]. The study was approved by the ethical committees of all institutions. The study was conducted in accordance with the declaration of Helsinki.

### 2.2. Pilot Study

Commercially available calibrators for the individual DOACs (Technoclone, Vienna, Austria) were tested in the anti-Xa assay using three different sample pre-dilutions (1:5, 1:10, and 1:20 in assay dilution buffer). All calibrators had known concentrations of the respective DOAC, which was traceable to the LC-MS/MS method of the developing company. In a second set of experiments, using the calibrators for apixaban and rivaroxaban, the edoxaban-containing calibrators were assigned values for the other DOACs. Using these assigned values, an anti-Xa calibration curve was established for apixaban and rivaroxaban using the edoxaban calibrators. Commercially available DOAC-specific controls were tested using the calibrated anti-Xa assay.

### 2.3. Data Collection and Handling of Samples

All data were anonymized and collected in a secured RedCAP database. The following data were recorded: sex, age, and drug used. To ensure appropriate pre-analytic conditions, detailed protocols were put in place at all institutions [21]. Venous blood samples were drawn in plastic syringes containing 1 ml trisodium citrate (0.106 mol L^−1^) per mL of blood. To avoid contamination with tissue factor, the citrated samples were collected after obtaining syringes containing different anticoagulants. The application of tourniquet was minimized and removed before transport and processing. Aliquots were frozen at −80 °C, kept frozen during a delivery time of 3 to 4 h (on dry ice) to Inselspital, central laboratory, and kept continuously without any freeze–thaw cycle. The samples were stored for 1 to 13 months before analysis.

### 2.4. Determination of the Edoxaban-Calibrated Anti-Xa Assay

Following the pilot study mentioned above, we selected the TECHNOCHROM^®^ anti-Xa assay using an edoxaban calibrator (Technoclone, Vienna, Austria). The Ceveron^®^ s100 analyzer was applied (Technoclone, Vienna, Austria). Samples were analyzed immediately after thawing (15 min; 37 °C) in a 1:20 dilution following the manufacturer’s instructions. Briefly, reagent 2 (bovine factor Xa) and reagent 3 (chromogenic substrate) were reconstituted separately in 4 mL distilled water and preheated to 37 °C. Reagent 1 consisted of 20 mL anti-Xa buffer and TRIS-EDTA buffer (pH 8.4). The samples were incubated in a cuvette of the analyzer kept at 37 °C (50 μL of the diluted sample, 50 μL of bovine factor Xa, 50 μL of Xa substrate). The kinetic absorbance was measured at 405 nm. Neither clinical information nor LC-MS/MS results were known to the performer of the anti-Xa assay.

### 2.5. Determination of LC-MS/MS

To quantify the concentrations of all anti-Xa inhibitors (apixaban, rivaroxaban, and edoxaban—including edoxaban metabolite M4), ultra-high-performance liquid chromatography-tandem mass spectrometry (LC-MS/MS) was conducted as a reference standard. The analytical procedures have been described previously [16,18,20]. Briefly, for analyte extraction and protein precipitation, 10 µL acetonitrile/water 1:1 (*v*/*v*), 25 µL extraction buffer (MassTox TDM Series A, Chromsystems Instruments & Chemicals GmbH, Gräfelfing, Germany), and 240 µL precipitation reagent (MassTox TDM Series A, Chromsystems Instruments & Chemicals GmbH, Gräfelfing, Germany) including the internal standards (apixaban 13C,D3, rivaroxaban 13C6, edoxaban 13C,D2) were added to 50 µL plasma. After vortexing, the samples were centrifuged at 14,000× *g* rcf and 20 °C for 4 min. An amount of 20 µL of the supernatant material was diluted with 80 µL of water/methanol 8:2 (*v*/*v*) and stored at 10 °C. Pooled plasma was used to prepare calibrators and QC (Innovative Research, Novi, MI, USA). Three microliters of the specimen were measured by reversed-phase chromatography (Cortecs UPLC C18 column, 2.1 × 75 mm, 1.7 µm, Waters, Milford, MA, USA) on a triple quadrupole mass spectrometer (Xevo TQ-S, Waters) coupled to a UPLC Acquity I-Class system (Waters). Rivaroxaban, apixaban, edoxaban, and edoxaban M4 were separated at 0.4 mL/min with a gradient using water (A) and methanol (B) acidified with 0.1% (*v*/*v*) formic acid (mobiles phase) (0.0–0.5 min; 20% B; 0.5–2.5 min, 20–99% B; 2.5–3.5 min, 99% B; 3.5–3.51 min, 99–20% B; 3.51–4.5 min, 20% B). We optimized source offset and transition parameters for each analyte. TargetLynx of the MassLynx software was used to process the raw data (version 4.1, Waters). For edoxaban analysis, edoxaban M4 metabolites were added to edoxaban. Apixaban, rivaroxaban, and edoxaban pure substances were provided by Bristol Myers Squibb Company (New Brunswick, NJ, USA), Bayer AG (Wuppertal, Germany), and Daichi Sankyo Co, Ltd. (Tokyo, Japan). The LC-MS/MS assay was validated according to international guidelines and fulfilled their acceptance criteria [22,23]. Inter-/intra-day imprecision and inaccuracy are given in Appendix A. Clinical information and index test results were not available to the performer of LC-MS/MS.

### 2.6. Statistical Analysis

Descriptive statistics (numbers/percentages or median/range) were used as appropriate. Spearman correlation coefficients (r_s_), Deming regressions, and Bland and Altman difference plots were used to describe the relationship between anti-Xa measurements and drug concentrations (LC-MS/MS). Correlation coefficients of 0.9 were considered accurate, and those of 0.6 were considered inadequate. Cut-off levels of the edoxaban-calibrated anti-Xa assay in relation to clinically relevant drug concentrations (30 mgL^−1^, 50 mgL^−1^, and 100 mgL^−1^) were established by receiver operating characteristics curve analysis (ROC). Two-by-two tables were created, and sensitivities and specificities were calculated. Analyses were performed using Stata 14.1 (Stata-Corp. 2015. Stata Statistical Software: Release 14. Stata-Corp LP, College Station, TX, USA). Prism 8 was used to generate all figures (GraphPad Software, Inc., La Jolla, CA, USA).

## 3. Results

### 3.1. Pilot Study

When plotting the raw data from calibrators for apixaban, edoxaban, and rivaroxaban in an anti-Xa assay, very similar patterns could be observed. Using different sample pre-dilutions typically employed in anti-Xa assays, the different DOACs behaved similarly, exhibiting a steep discrepancy between 0 and approx. 100 µgL^−1^ using a low sample dilution (e.g., 1:5) and a broad variation between 50 and 500 µgL^−1^ using a high sample dilution (e.g., 1:20).

In a small calibration exercise, ten samples of the edoxaban calibrator set were assigned values for apixaban and rivaroxaban using a calibrated anti-Xa assay (Table 1).

Then, using these newly assigned values, DOAC-specific calibration curves were generated, and quality control samples for apixaban and rivaroxaban were measured. All controls recovered within the acceptable ranges from the manufacturer.

### 3.2. Patient Characteristics

Eventually, 932 patients were included from nine study centers (Figure 1). We excluded 375 for several reasons: additional application of unfractionated or low-molecular-weight heparin (*n* = 35), more than one drug was detected (*n* = 2), pre-analytical interfering factors (*n* = 5), and not enough specimen material (*n* = 337). Overall, 553 were considered for the present analysis. In total, 221 patients were treated with apixban (40.0%), 300 with rivaroxaban (54.3%), and 32 with edoxaban (5.7%). The median age was 74 years (IQR 63 to 83), and 40.1% of the individuals were female. Detailed characteristics can be found in Table 2.

### 3.3. Accuracy of the Edoxaban-Calibrated Anti-Xa Assay

The associations between edoxaban-calibrated anti-Xa results and apixaban, rivaroxaban, and edoxaban drug concentrations are shown in Figure 2. The correlation coefficient (r_s_) of all measurements was 0.94 (95% confidence interval [CI] 0.93, 0.95). It was 0.95 for individuals treated with rivaroxaban (95% CI 0.94, 0.96), 0.93 with apixaban (95% CI 0.91, 0.95), and 0.93 with edoxaban (95% CI 0.86, 0.97). The coefficients of the regression equations (slope and Y-intercept) are reported in Table 3.

### 3.4. Detection of Clinically Significant Drug Levels

Figure 3 shows the distribution of edoxaban-calibrated anti-Xa measurements in individuals with and without clinically relevant drug concentrations (30 µgL^−1^, 50 µgL^−1^, and 100 µgL^−1^, respectively). The optimal cut-off values derived from receiver operating characteristics (ROC) analysis were 34 µg/L^−1^ for 30 µgL^−1^, 49 µg/L^−1^ for 50 µgL^−1^, and 83 µg/L^−1^ for 100 µgL^−1^. The performance in terms of areas under the ROC curve was 0.96, 0.98, and 0.99, respectively (Figure 4). The edoxaban-calibrated anti-Xa assay detected clinically relevant drug concentrations with sensitivities of 92.3% (30 µgL^−1^; 95% CI 89.2, 94.6), 92.7% (50 µgL^−1^; 95% CI 89.4, 95.1), and 94.8% (100 µgL^−1^; 95% CI 91.1, 97.0). The specificity was 82.2% (95% CI 75.6, 87.3), 93.7% (95% CI 89.7, 96.2), and 94.4% (95% CI 91.3, 96.4).

## 4. Discussion

Using a specimen of 553 patients enrolled in a large prospective study, we assessed the clinical performance of an edoxaban-calibrated anti-Xa assay to determine critical apixaban, rivaroxaban, or edoxaban drug concentrations in clinical practice. All drugs showed solid accuracy concerning correlation coefficients (r_s_) and areas under ROC curves. In addition, clinically relevant drug concentrations were captured with a high sensitivity and specificity.

These results align with previous studies of various study designs using other reagents. In different analyses of the same study, we demonstrated recently that both a low-molecular-weight heparin-calibrated anti-Xa assay [16] and an assay calibrated to unfractionated heparin can accurately measure apixaban, rivaroxaban, and edoxaban drug concentrations [18]. High accuracy and consistency of a rivaroxaban-calibrated anti-Xa assay with rivaroxaban plasma concentrations was demonstrated in 20 healthy individuals [17]. A cross-sectional study analyzing samples of 30 patients taking rivaroxaban showed a high correlation between LC-MS/MS and heparin anti-factor Xa activity [24]. A strong correlation between heparin-calibrated anti-Xa activity and LC-MS/MS measurements was also found in a retrospective cohort of 24 patients taking rivaroxaban or apixaban [25]. Moreover, 241 left-over specimens were used to determine the correlation between UFH-calibrated and rivaroxaban-/apixaban-calibrated anti-Xa measurements, concluding that there was a high correlation. Several additional studies assessed anti-Xa results’ association with LC-MS/MS measurements in spiked samples [26,27,28,29].

An important strength of our investigation was the large number of patients analyzed, which increased precision and statistical power. As a multicenter study conducted in clinical practice, the study adequately reflects the potential clinical application of the test. LC-MS/MS was also used as a reference standard, considered the most appropriate method for measuring DOAC [16,30]. However, our study is associated with limitations as well. First, only one edoxaban-calibrated reagent was tested, and we cannot be sure that other assays behave similarly. Thus, our results must be repeated in other settings using other reagents as well. In addition, fewer individuals received edoxaban treatment compared to rivaroxaban and edoxaban. We believe that the large sample size compensated for this, however.

Our results suggest that an edoxaban-calibrated anti-Xa assay is a feasible laboratory test to detect clinically relevant DOAC concentrations in clinical practice. The most straightforward way to apply this test in clinical practice would be to use the above-mentioned cut-off values of 30, 50, and 100 µgL^−1^. As an alternative, the actual values could also be calculated with the help of regression equations. However, caution must be taken here, as there is a certain degree of dispersion. Of note, other reagents and tests must be evaluated in additional studies before implementation. One might argue that the edoxaban-calibrated anti-Xa test could also be used to monitor DOAC treatment. However, this is a controversial topic, and no clinical guideline currently recommends this outside of clinical trials. The present study aimed to assess the diagnostic accuracy of an anti-Xa assay with regard to clinically relevant cut-off concentrations, that support decision making with regard to (a) reversal agents in case of acute bleeding, (b) scheduling urgent surgery, and (c) thrombolysis in case of acute stroke. We believe that the current study does not support the monitoring of DOAC with the help of uni-calibrated anti-Xa assays.

## 5. Conclusions

In a large prospective clinical study including 553 patients treated in real-life clinical practice, the accuracy of an edoxaban-calibrated anti-Xa assay with apixaban, rivaroxaban, and edoxaban drug levels was solid and most clinically relevant drug concentrations were predicted correctly. However, the accuracy of other edoxaban-calibrated reagents must be confirmed in future studies.

## Figures and Tables

**Figure 1 diagnostics-13-02128-f001:**
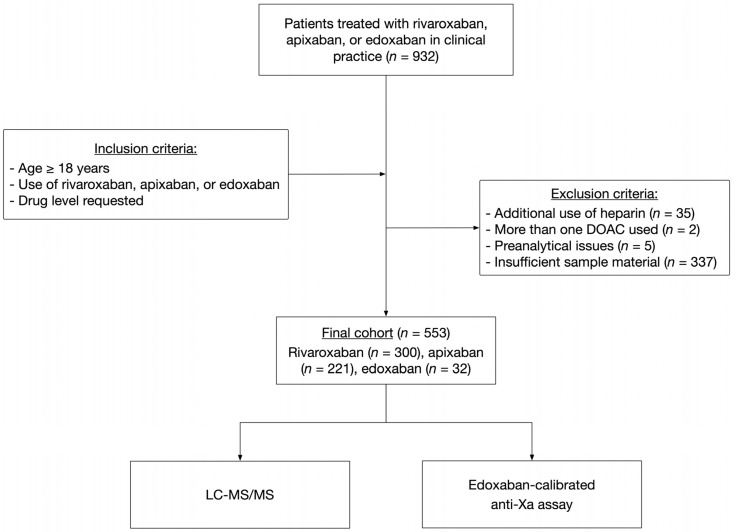
The flow of the patients.

**Figure 2 diagnostics-13-02128-f002:**
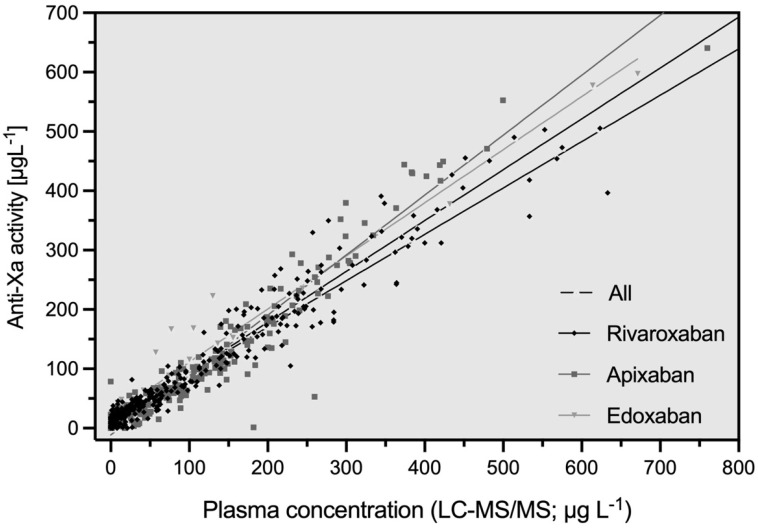
Scattergram showing the association between edoxaban-calibrated anti-Xa activity and the measurements of LC-MS/MS by drug. The overall correlation (r_s_) was 0.94 (95% CI 0.93, 0.95). It was 0.95 for rivaroxaban (95% CI 0.94, 0.96), 0.93 for apixaban (95% CI 0.91, 0.95), and 0.93 for edoxaban (95% CI 0.86, 0.97).

**Figure 3 diagnostics-13-02128-f003:**
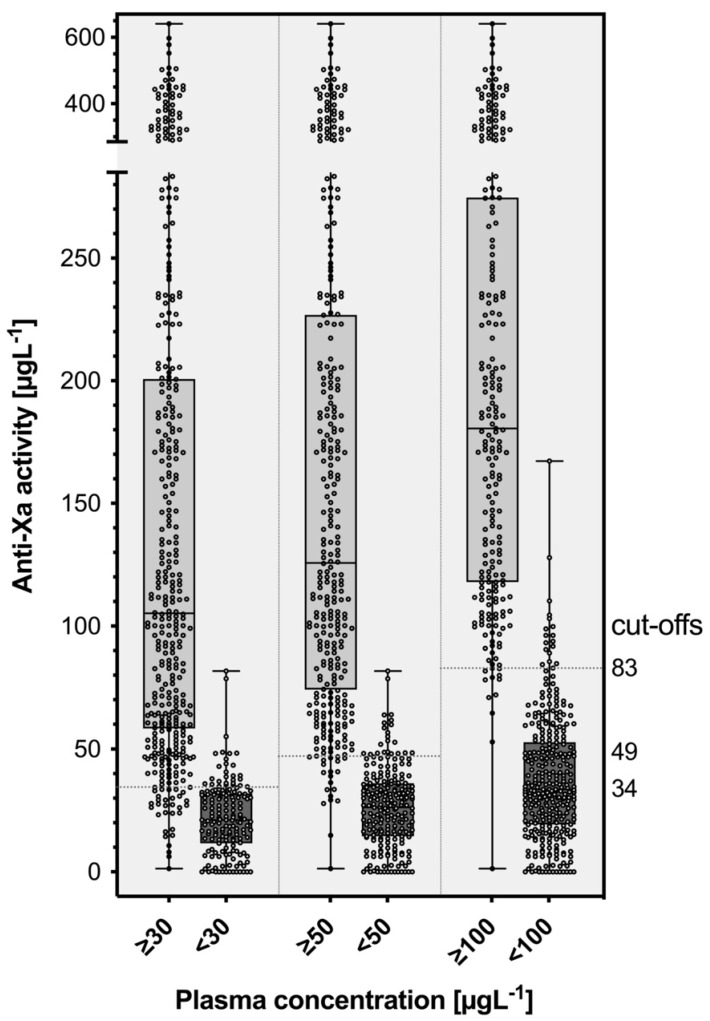
Distribution of edoxaban-calibrated anti-Xa measurements in patients with and without clinically relevant drug concentrations (30 µgL^−1^, 50 µgL^−1^, 100 µgL^−1^). The cut-off thresholds were determined using a receiver operating characteristics curve (ROC) analysis. The sensitivities were 92.3% (30 µgL^−1^; 95% CI 89.2, 94.6), 92.7% (50 µgL^−1^; 95% CI 89.4, 95.1), and 94.8% (100 µgL^−1^; 95% CI 91.1, 97.0). The specificities were 82.2% (95% CI 75.6, 87.3), 93.7% (95% CI 89.7, 96.2), and 94.4% (95% CI 91.3, 96.4).

**Figure 4 diagnostics-13-02128-f004:**
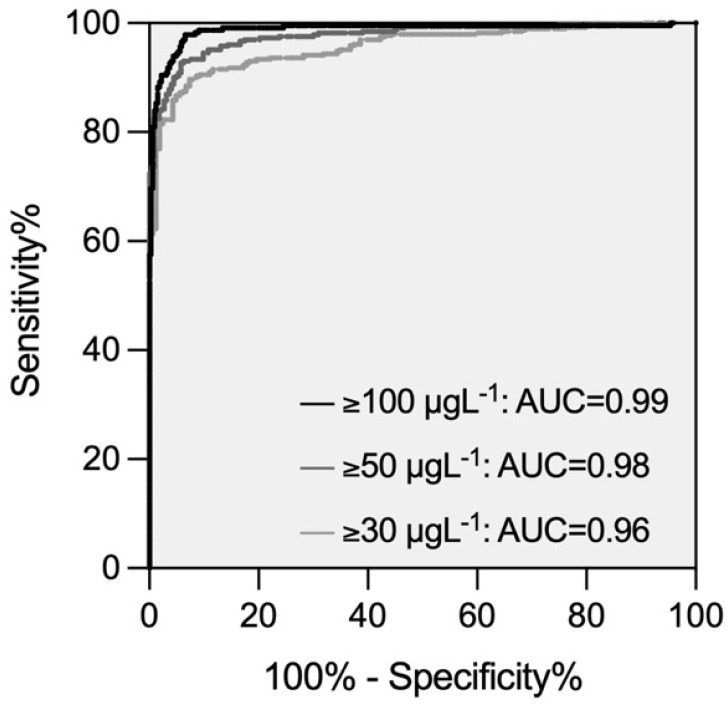
Receiver operating characteristics curves demonstrating the diagnostic accuracy of an edoxaban-calibrated anti-Xa assay for the prediction of clinically relevant drug concentrations (30 µgL^−1^, 50 µgL^−1^, 100 µgL^−1^).

**Table 1 diagnostics-13-02128-t001:** Results of a pilot study assigning values for apixaban and rivaroxaban to an edoxaban calibrator set.

Calibrator	Apixaban (µgL^−1^)	Edoxaban (µgL^−1^)	Rivaroxaban (µgL^−1^)
CAL 1	0	0	0
CAL 2	26	50	38
CAL 3	119	144	124
CAL 4	255	316	234
CAL 5	447	506	385

**Table 2 diagnostics-13-02128-t002:** Patient characteristics (*n* = 553).

	Rivaroxaban	Apixaban	Edoxaban	All
**Patients** (*n*/%)	300 (54.3)	221 (40.0)	32 (5.7)	553 (100)
**Age** (median/IQR)	74 (63–83)	78 (67–82)	75 (58–81)	76 (65–82)
**Female sex** (*n*/%)	126 (40.1)	86 (39.8)	10 (32.3)	222 (40.1)

**Table 3 diagnostics-13-02128-t003:** Accuracy of an edoxaban-calibrated anti-Xa assay for measuring all anti-Xa inhibitors: apixaban, rivaroxaban, and edoxaban. Patients treated in clinical practice were enrolled in a multicenter cross-sectional study (*n* = 553). An LC-MS/MS analysis was performed as a reference standard.

	Rivaroxaban	Apixaban	Edoxaban	All
**Spearman’s correlation coefficient** (95% CI)	0.95 (0.94, 0.96)	0.93 (0.91, 0.95)	0.93 (0.86, 0.97)	0.94 (0.93, 0.95)
**Deming regression** **slope** (95% CI)	0.78 (0.70, 0.87)	1.01 (0.94, 1.08)	0.90 (0.83, 0.96)	0.86 (0.79, 0.92)
**Y-Intercept**	14.08 (6.28, 21.87)	−11.39 (−19.25, −3.53)	21.73 (10.69, 32.77)	6.97 (0.74, 13.19)

## Data Availability

The dataset can be requested from the authors upon reasonable request.

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
