# Peer review of "Determination of Anti-Xa Inhibitor Plasma Concentrations Using a Universal Edoxaban Calibrator"

_diagnostics, 2023, doi:10.3390/diagnostics13122128_

Round 1

Reviewer 1 Report

 Thank you for allowing me to review your manuscript.

While the overall assay outcomes and need are clearly important and relatable to the clinic, there are several parts of this manuscript that gloss over and do not provide sufficient detail on the outputs being compared. In addition, several intermediate steps are not presented in graphical or tabular form so readers cannot independently assess the validity of claims of the assay. For instance, It is curious why common accuracy and precision characteristics of the LC-MS and Thechnocrom assays were not discusses (both independently and then when compared before determining the Spearman coefficient). The limitations also need to be expanded to include thoughtfulness for any potential future clinical application.

General Comments:

** While stated as a limitation, it is not great that only one lot of calibrator was tested. Three independent lots should have been evaluated. Being sympathetic and knowing the authors realize as such, the conclusions should be plainly stated that this is a ‘genalized proof of concept’ and not qualification or validation of their ROC or assay sensitivity/specificity.

** No discussion of inter- or intra-assay precision and accuracy. It is hard to determine the reliability of the LC-MS quantification as well as the coagulation-styles assay without assessments, especially as it is also being carried out on non-IVD equipment/controlled.

** The non-LC-MS equipment being used is RUO and there is no discussion of instrument controls/calibration/quality control to ensure machines are operating consistently and reliably. You can use standard clinical lab for flow cytometers as an example.

** While anti-Xa activity cut-points were determined for the ROC analysis, there is no discussion on the clinical applicability of either the plasma concentrations or activity cut-offs concluded in this pilot study versus the extensive literature and clinical experience (i.e., why are there assay driven cut-offs, but no discussion of the clinically relevant ones?).

Specific Comments:

** Based on sample handling, it seems that the Thechnocrom uses diluted whole blood versus plasma in the LC-MS analysis. Please clarify and state clearly.

** Section 2.4: It should be stated clearly the type of instrument and output of the Ceveron s100 analyzer so it is more clear to a reader the outputs being compared and how they are obtained.

** Table 1: It is this reviewers personal suggestion that a graphical figure be presented alongside the table to allow the reader to also assess if the assigned values are sensical and appropriate.

** Table3: While I understand where the authros are going, the coefficient and line characteristics come after the fact of fitting. Accuracy needs to show LC-MS measurements (reference) of drug compared to your Thechnocrom assay (test) fall within an acceptable criteria, not a generalized population fit. SE should also be provided for rpelicates (or if no replicates, state as limitation for the assay qualification and comparison).

** Table 3: Why 96% CI? Or is this a typo.

General copy editing should suffice. 

Author Response

On behalf of all co-authors, we would like to thank you for taking the time to carefully review our manuscript. We feel that the reviewers’ comments have helped to improve the quality of the study substantially. Please find attached a revised version with marked changes as well as a clean version of the manuscript. Point-by-point responses are given below.

** While stated as a limitation, it is not great that only one lot of calibrator was tested. Three independent lots should have been evaluated. Being sympathetic and knowing the authors realize as such, the conclusions should be plainly stated that this is a ‘genalized proof of concept’ and not qualification or validation of their ROC or assay sensitivity/specificity.

RESPONSE: We fully agree with the reviewer that the aim of the present study was not to prove that the TECHNOCHROM® anti-Xa assay accurately determines DOAC drug concentrations. These data can be accessed in the manufacturer's accompanying documents and in various previous publications. The present study is indeed a proof-of-principle study to determine the diagnostic accuracy of an anti-Xa assay utilizing a universal edoxaban calibrator to determine critical plasma concentrations of all anti-Xa inhibitors (rivaroxaban, edoxaban, and apixaban). We added some comments in the introduction section (line 82 to line 85) and the discussion section (line 292-295)

** No discussion of inter- or intra-assay precision and accuracy. It is hard to determine the reliability of the LC-MS quantification as well as the coagulation-styles assay without assessments, especially as it is also being carried out on non-IVD equipment/controlled.

RESPONSE: We thank the reviewer for pointing out this important point. The LC-MS/MS assay was validated according to international guidelines. A sentence was added in the method section for clarifying this point (line 163 to line 165). The inter-/intra-day imprecision and inaccuracy results obtained for this assay were added in appendix (see table S1 of the supplementary material).

** The non-LC-MS equipment being used is RUO and there is no discussion of instrument controls/calibration/quality control to ensure machines are operating consistently and reliably. You can use standard clinical lab for flow cytometers as an example.

RESPONSE: Thank you for the possibility to clarify this point. The analyzer used was a standard commercial coagulation analyzer which is similar to others available. Reagents and analyzer are are approved IVD products in the European Union and all required quality criteria have been raised and fulfilled. Calibrator and controls are described and are typically assays specific. In this particular study, an edoxaban calibrator set has been used with additional values for apixaban and rivaroxaban. We refer to the paragraph line 127 to line 138.

** While anti-Xa activity cut-points were determined for the ROC analysis, there is no discussion on the clinical applicability of either the plasma concentrations or activity cut-offs concluded in this pilot study versus the extensive literature and clinical experience (i.e., why are there assay driven cut-offs, but no discussion of the clinically relevant ones?).

RESPONSE: Thank you for pointing out this important point. The main aim was to determine the diagnostic accuracy with regard to critical clinical plasma concentrations of rivaroxaban, apixaban, and edoxaban (line 82 to line 85). Following recommendations of the International Society on Thrombosis and Haemostasis (ISTH), we considered 30 mgL-1, 50 mgL-1, and 100 mgL-1 to be clinically relevant (line 99 to line 100). These concentrations correspond with a higher perioperative bleeding risk in case of (a) high-risk surgery, (b) normal risk surgery, and (c) thrombolysis in patients with acute stroke (line 292 to line 295). The corresponding cut-off levels of the anti-Xa test was determined following using ROC-curve (line 173 to line 176). Finally, 4x4 tables were created and the sensitivity/ specificity calculated using the category of the anti-Xa test and the LS-MS/MS measurements (line 176).

** Based on sample handling, it seems that the Thechnocrom uses diluted whole blood versus plasma in the LC-MS analysis. Please clarify and state clearly.

RESPONSE: Thank you for raising this important point. Venous blood samples were drawn in plastic syringes containing 1ml trisodium citrate (0.106 mol L-1) per mL of blood (line 116 to line 118).

** Section 2.4: It should be stated clearly the type of instrument and output of the Ceveron s100 analyzer so it is more clear to a reader the outputs being compared and how they are obtained.

RESPONSE: Thank you for the possibility to clarify this point. The Assay principle of chromogenic antiXa has been well established and is widely used in clinical practice and scientific inquiry. Briefly, DOACs are oral factor Xa inhibitors, that selectively and directly inhibit factor Xa. Anti Xa reagent, with a constant amount of FXa, is added to the sample. Part of the added FXa is inhibited by the respective DOAC, while the remaining FXa cleaves the chromogenic Xa substrate Suc Ile Glu(piperidin 1 yl) Gly Arg pNA·HCl. Formation of p nitroaniline, measured at 405 nm, is inversely proportional to the amount of DOAC in the sample (line 126 to line 138).

** Table 1: It is this reviewers personal suggestion that a graphical figure be presented alongside the table to allow the reader to also assess if the assigned values are sensical and appropriate.

RESPONSE: We thank the reviewer for this proposal. To follow this we created several graphs. When we then compared them to the already existing figures 2 and 3, we found that the information content is much lower. Since this clearly disturbed the consistency of the manuscript, we finally decided not to use them. I hope that the reviewer can still follow us here.

** Table3: While I understand where the authros are going, the coefficient and line characteristics come after the fact of fitting. Accuracy needs to show LC-MS measurements (reference) of drug compared to your Thechnocrom assay (test) fall within an acceptable criteria, not a generalized population fit. SE should also be provided for rpelicates (or if no replicates, state as limitation for the assay qualification and comparison).

RESPONSE: Thank you for raising this point. The primary aim of the study was to determine the diagnostic accuracy with regard to critical clinical plasma concentrations of rivaroxaban, apixaban, and edoxaban. As such, this is a diagnostic accuracy study that has to follow current guidelines on diagnostic accuracy studies, the STARD Guideline in particular (e.g. https://doi.org/10.1373/clinchem.2015.246280). In addition, we followed the conventions used in diagnostic accuracy studies in the field of hemostasis and monitoring of anticoagulation. The corresponding author has himself performed numerous such studies, some of which have been published in high-impact journals such as Blood, the British Journal of Haematology, and the Journal of Thrombosis and Haemostasis (https://pubmed.ncbi.nlm.nih.gov/collections/58103594/?sort=pubdate). We hope that the reviewers will follow us here. 

** Table 3: Why 96% CI? Or is this a typo.

RESPONSE: Thank you for spotting this typing error. We corrected it accordingly in Table 3.

Reviewer 2 Report

In this manuscript, Burger et al. assessed the clinical performance of an edoxaban-calibrated anti-Xa assay to determine apixaban, rivaroxaban, or edoxaban drug concentrations in clinical practice.

Overall, this is a well-structured and well-described paper. The submitted manuscript strictly complies with the task it pursues and is suitable for publication.

I don’t have to propose substantial corrections, however, some minor revisions could improve the manuscript:

- Please check typos and spacing;

- It should be interesting to insert a discussion about the importance of direct and indirect tests to determine the NOACs concentration. Post-marketing studies and meta-analyses have highlighted the potential occurrence of unpredictable safety signals. There are concerns and safety issues in real-world practice. Like vitamin K antagonists, NOACs showed bleeding as the most common adverse reaction, including gastrointestinal bleeding (GIB): it was described a higher risk of GIB in patients treated with high-dose of NOACs compared to those treated with warfarin. Beyond bleeding complications, different systematic reviews and meta-analyses assessed the potential risk of myocardial infarction, liver and renal injury, and nervous system disorders. Liver injury risk, as the rare case of jaundice due to intrahepatic cholestasis, is a recent safety issue that was undetected in clinical phases and emerged only from post-marketing analysis, especially for rivaroxaban. Not only patients with chronic kidney disease, but other patients as well, could have a higher risk of developing acute kidney injury if the therapeutic range of anticoagulation is exceeded. All these cases underline the necessity of careful post-marketing surveillance, considering the rapidly increasing number of patients treated with NOACs and patient’s risk factors such as old age, high polypharmacy rate, co-morbidities, and peculiar genetic background related to NOACs pharmacokinetic features.

.

Minor editing of English language required

Author Response

On behalf of all co-authors, we would like to thank you for taking the time to carefully review our manuscript. We feel that the reviewers’ comments have helped to improve the quality of the study substantially. Please find attached a revised version with marked changes as well as a clean version of the manuscript. Point-by-point responses are given below.

- Please check typos and spacing

RESPONSE: Thank you for spotting. We checked and corrected typos and spacing throughout the manuscript.

- It should be interesting to insert a discussion about the importance of direct and indirect tests to determine the NOACs concentration. Post-marketing studies and meta-analyses have highlighted the potential occurrence of unpredictable safety signals. There are concerns and safety issues in real-world practice. Like vitamin K antagonists, NOACs showed bleeding as the most common adverse reaction, including gastrointestinal bleeding (GIB): it was described a higher risk of GIB in patients treated with high-dose of NOACs compared to those treated with warfarin. Beyond bleeding complications, different systematic reviews and meta-analyses assessed the potential risk of myocardial infarction, liver and renal injury, and nervous system disorders. Liver injury risk, as the rare case of jaundice due to intrahepatic cholestasis, is a recent safety issue that was undetected in clinical phases and emerged only from post-marketing analysis, especially for rivaroxaban. Not only patients with chronic kidney disease, but other patients as well, could have a higher risk of developing acute kidney injury if the therapeutic range of anticoagulation is exceeded. All these cases underline the necessity of careful post-marketing surveillance, considering the rapidly increasing number of patients treated with NOACs and patient’s risk factors such as old age, high polypharmacy rate, co-morbidities, and peculiar genetic background related to NOACs pharmacokinetic features.

RESPONSE: Thank you for raising this important issue. One might indeed argue that the edoxaban-calibrated anti-Xa test could also be used to monitor DOAC treatment. However, this is a controversial topic, and no clinical guideline currently recommends this outside of clinical trials. The present study aimed to assess the diagnostic accuracy of an anti-Xa assay with regard to clinically relevant cut-off concentrations, that support decision making with regard to (a) reversal agents in case of acute bleeding, (b) scheduling urgent surgery, and (c) thrombolysis in case of acute stroke. We believe, however, that the current study does not support the monitoring of DOAC with the help of a uni-calibrated anti-Xa assays because estimating precise drug concentrations is not easy. We have added a comment on line 289 to line 297 to make this clear.